# Ultra-large single crystals by abnormal grain growth

Tomoe Kusama[1], Toshihiro Omori[1], Takashi Saito[1], Sumio Kise[2], Toyonobu Tanaka[2], Yoshikazu Araki[3] & Ryosuke Kainuma[1]

Producing a single crystal is expensive because of low mass productivity. Therefore, many metallic materials are being used in polycrystalline form, even though material properties are superior in a single crystal. Here we show that an extraordinarily large Cu-Al-Mn single crystal can be obtained by abnormal grain growth (AGG) induced by simple heat treatment with high mass productivity. In AGG, the sub-boundary energy introduced by cyclic heat treatment (CHT) is dominant in the driving pressure, and the grain boundary migration rate is accelerated by repeating the low-temperature CHT due to the increase of the sub-boundary energy. With such treatment, fabrication of single crystal bars 70 cm in length is achieved. This result ensures that the range of applications of shape memory alloys will spread beyond small-sized devices to large-scale components and may enable new applications of single crystals in other metallic and ceramics materials having similar microstructural features.

[1] Department of Materials Science, Graduate School of Engineering, Tohoku University, Aoba-yama 6-6-02, Sendai, Miyagi 980-8579, Japan. [2] Technology Development Department, Special Metals Division, Furukawa Techno Material Co., Ltd., 5-1-8 Higashi-yawata, Hiratsuka, Kanagawa 254-0016, Japan. [3] Department of Architecture and Architectural Engineering, Graduate School of Engineering, Kyoto University, Katsura, Nishikyo, Kyoto 615-8540, Japan. Correspondence and requests for materials should be addressed to T.O. (email: omori@material.tohoku.ac.jp)

Most metals, semiconductors and ceramics are composed of a large number of crystals, i.e., polycrystalline structure. Although some kinds of materials including not only semiconductors and ceramics but also shape memory alloys[1–6] and heat-resistant alloys[7–9] show superior properties in single crystalline form, the use of single crystals is, however, restricted to some special applications due to the high cost of processing. Single crystals are normally fabricated by crystal growth methods during solidification, such as the Bridgman process and the Czochralski processes[10–12]. Another possibility for single crystal production is a solid-state technique utilizing abnormal grain growth (AGG). In polycrystalline materials, grain growth occurs to reduce the fraction of grain boundaries (GBs) with high energy, in which the grain structure coarsens by gradual growth of larger grains and elimination of smaller ones[13]. The distribution of grain size is relatively uniform during normal grain growth (NGG). In certain circumstances, only limited grains consume the surrounding smaller grains and grow rapidly, which is called AGG[13–15]. Several methods to obtain a single crystal using AGG in a solid state have been reported, one of the most well-known methods being the strain-anneal method using slight cold-deformation followed by thermal annealing[16, 17].

Very recently, the present authors reported a new AGG phenomenon in the β (bcc: body centered cubic) phase induced by cyclic heat treatment (CHT) through β/β + α (fcc: face centered cubic) phase transformation in Cu-Al-Mn shape memory alloy[18]. This technique is highly advantageous for obtaining a single crystal. In general, when abnormally growing

**Fig. 1** Single crystal and oligocrystalline Cu-Al-Mn shape memory alloy bars. **a** Diagram of cyclic heat treatment (CHT) with a combination of the high-temperature cycle (HTC) and low-temperature cycle (LTC) for obtaining a single crystal (WQ: water quenching). **b** Cu-Al-Mn single crystal bars 15 mm in diameter and 700 mm in length obtained by the CHT shown in Fig. 1a, and (001) pole figures at the both ends of the bar, which show that the crystal orientations at the both ends are almost the same. *Scale bar*, 50 mm. **c** Diagram of HTC without LTC. **d** Cu-Al-Mn alloy bar with a bamboo structure 15 mm in diameter and 700 mm in length, subjected to CHT shown in Fig. 1c. The maximum length of a crystallographic grain is about 250 mm and some grain boundaries, as indicated by arrows, are always detected in the bar specimens after this treatment. *Scale bar*, 50 mm

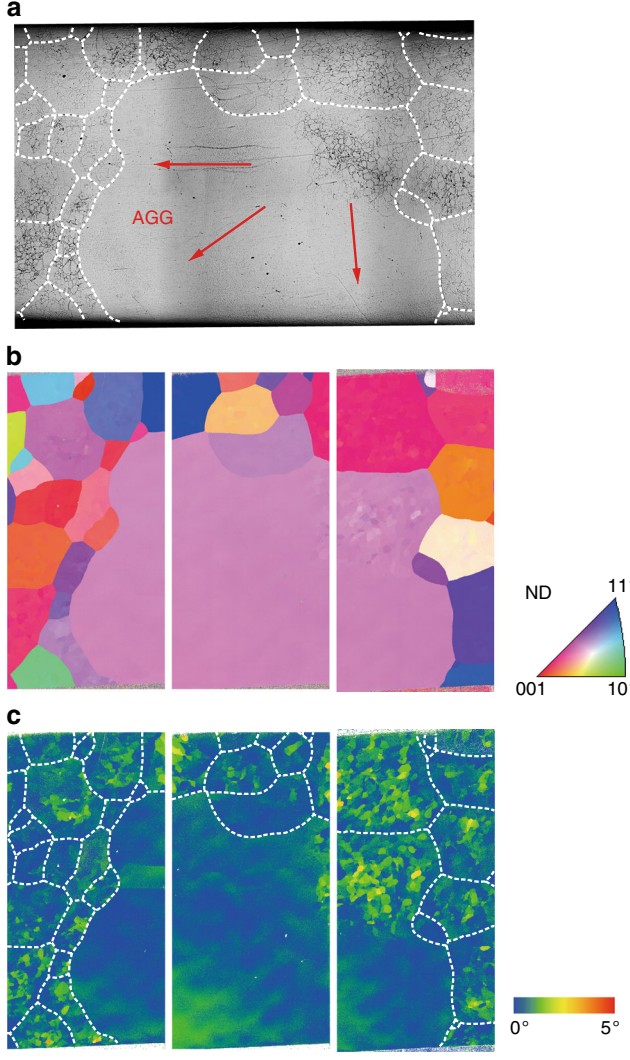

**Fig. 2** Microstructure of Cu-Al-Mn alloy during abnormal grain growth (AGG). **a** Optical micrograph of Cu-Al-Mn sheet quenched from 900 °C in the heating process after cooling from 900 to 500 °C (Supplementary Fig. 4a). The dashed lines indicate high-angle boundaries. **b** Inverse pole figure (IPF) mapping. The color means the crystal direction with respect to the normal direction of the sheet and corresponds to crystal direction given in the stereographic triangle. **c** Grain reference orientation deviation (GROD) mapping calculated as the misorientation angle with respect to a reference orientation (average orientation of a grain) in each grain. The dashed lines indicate high-angle boundaries. *Scale bar*, 1 mm

grains consume other small grains, the distribution of grain size finally reverts from bimodal to unimodal due to the contact with neighboring abnormal grains and the growth mode becomes NGG[13]. However, in the new technique, AGG can additionally and continuously occur after every CHT. We have previously reported a large grain 50 mm in length in a Cu-Al-Mn sheet obtained by repeating this AGG phenomenon caused by the normal heat cycle between 900 and 500 °C[18]. The origin of the AGG is still under discussion, but it has been pointed out that the subgrain structure, which is formed in the β matrix surrounding α precipitates during slow cooling from the β single-phase state (e.g., 900 °C) to the α + β two-phase state (e.g., 500 °C), plays an important role[18].

Recently, uses of shape memory alloys for seismic devices such as dampers and isolators have attracted considerable attention

since they can dissipate energy by stress-strain hysteresis, recover deformation upon unloading and limit force transmission[19, 20]. Ti-Ni shape memory alloy bars showing self-centering capability due to superelasticity are being used on a trial basis as a part of bridge columns to reduce permanent deformation by earthquakes[21]. However, the low machinability and cold-workability of the conventional Ti-Ni alloy are obstacles to its widespread use. Since Cu-Al-Mn shape memory alloys have high machinability and cold-workability[22], their application to seismic devices has been investigated[23–25]. The superelasticity of Cu-based shape memory alloys is drastically enhanced by increasing the grain size relative to the cross-sectional size of materials[1–3], and in particular, an ideal superelastic response can be obtained in a single crystal[26–28]. If the fabrication of large single crystal parts can be realized by simple heat-treatment, applications to seismic devices are expected to increase.

In the present study, the mechanism of the AGG induced by CHT in a Cu-Al-Mn alloy is investigated by microstructural analysis. GBs migrate consuming the subgrains formed during the CHT, leading to AGG, and the growth rate increases with increasing the misorientation between the subgrains. These facts suggest that the sub-boundary energy is a driving pressure of AGG, which is supported by the thermodynamic analysis. Based on this mechanism, a heating/cooling process, including low-temperature CHT between 740 and 500 °C, is developed for accelerated AGG, and as a result, fabrication of single crystal bars 700 mm in length is achieved by only CHT.

## Results

**Grain growth to single crystal**. In the present $Cu_{71.6}Al_{17}Mn_{11.4}$ (at%) alloy, the β single phase is stable at temperatures higher than 726 °C and the α-phase precipitates at lower temperatures[18, 29]. We conducted five heating/cooling cycles between 900 and 500 °C (high-temperature cycle: HTC), four cycles between 740 and 500 °C (low-temperature cycle: LTC), and final heating to 900 °C, as shown in Fig. 1a. Figure 1b shows Cu-Al-Mn bars with a dimension of 15 mmϕ × 700 mm subjected to the CHT of Fig. 1a. No GB is observed (except sub-boundaries) in the bars and both ends have almost the same orientation, as shown in the (001) pole figure, meaning that a single crystal 700 mm in length was obtained by the combination of HTC and LTC. On the other hand, in the Cu-Al-Mn bar obtained when skipping the LTC treatment (Fig. 1c), several GBs always remained, as shown in Fig. 1d.

**Microstructures**. In order to understand this AGG phenomenon, we investigated the microstructure of the Cu-Al-Mn alloy after CHT. Figure 2a–c shows an optical micrograph, inverse pole figure (IPF) mapping and grain reference orientation deviation (GROD) mapping, respectively, in a Cu-Al-Mn alloy immediately quenched from 900 °C without annealing after one HTC. As shown in the micrograph and mappings, abnormal grains (AGs) are surrounded by smaller grains containing a high density of subgrains with orientation deviation within ~3° to the neighboring subgrains. It is important to note that in the AG as well, the subgrain structure is locally observed in the upper-right region. A similar microstructure was detected in many other AGs in the alloy. This fact strongly suggests that the GBs of the AGs migrate, sweeping out the surrounding grains with subgrains, as indicated by red arrows in Fig. 2a.

This microstructural evolution is depicted in the schematic illustration of Fig. 3a. All the β-grains initially contain some density of subgrains which were introduced by the precipitation of the α-phase, as reported in our previous paper[18], and the pre-existing GBs are in contact with the subgrains on both sides

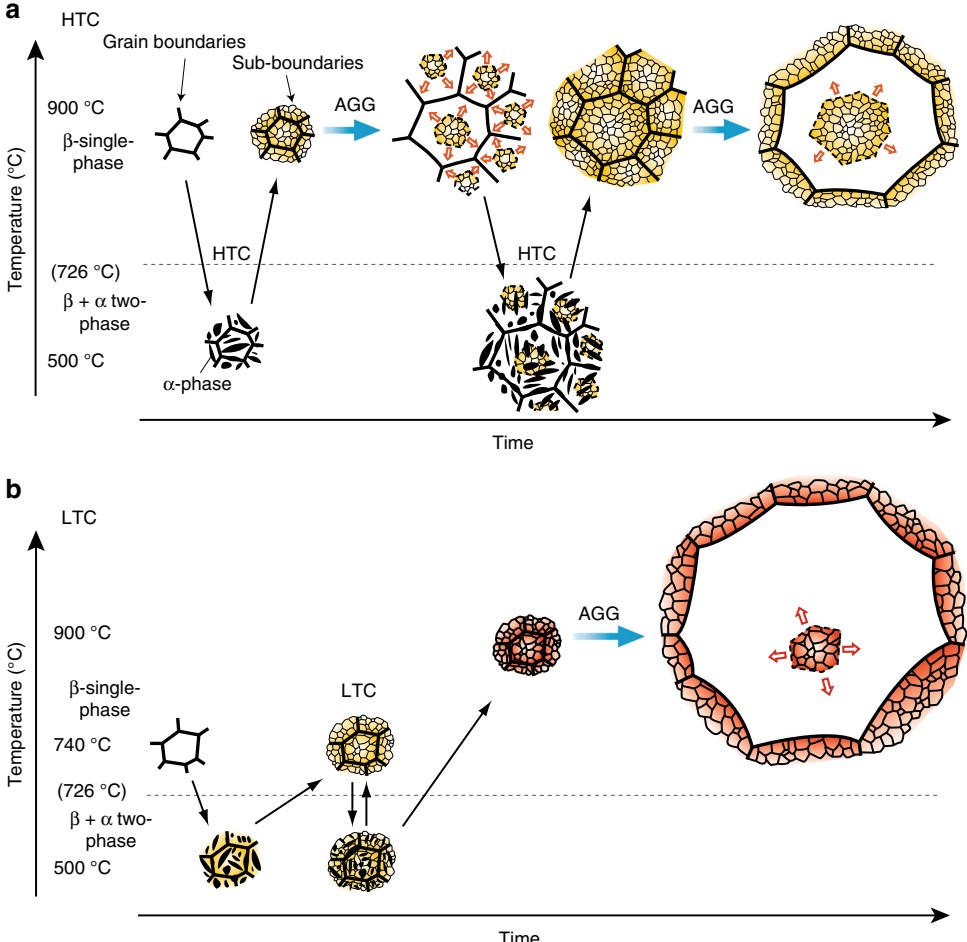

**Fig. 3** Schematic illustrations of abnormal grain growth (AGG) phenomenon. **a** In high-temperature cycles (HTC) (900/500 °C), the subgrain structure formed in association with α-phase precipitation at 500 °C remains after resolution of α-phase in heating. At 900 °C after HTC, some grains start AGG by consuming the surrounding subgrains, and this continues until the abnormal grain (AG) comes in contact with a neighboring AG, where the dominant driving pressure for grain boundary (GB) migration is the sub-boundary energy. AGG can repeatedly occur if HTC is repeated. **b** In multiple low-temperature cycles (LTCs) (740/500 °C), the sub-boundary energy increases due to the increasing misorientation between subgrains. As a result, the GB migration rate at 900 °C becomes faster and a super-large crystal can be realized

in the initial stage. When one grain occasionally starts to grow as NGG (Supplementary Discussion), a subgrain-free zone is formed only behind the moving GB. Since the GBs migrate consuming the sub-boundaries existing only in front of the moving GB, the growth rate of the grain becomes faster and the normal grain (NG) may change to an AG. That is, the boundary energy of subgrains may be one component of the driving pressure in the present AGG phenomenon. When the AGs growing in several regions come in contact with one another, AGG may be temporarily arrested due to the loss of driving pressure. The subgrain structure, however, can be restored by further CHT and AGG recommences. Thus, AGG can additionally be induced by repeating CHT and a large crystal can be consequently obtained.

**Grain boundary migration rate**. To quantitatively discuss this proposed mechanism of AGG, we experimentally estimated the growth rate of AGs, which encroached on the subgrains in some Cu-Al-Mn alloy sheets with different subgrain microstructures. The first set of sheets was cooled from 800 to 500 °C and then heated to 800 °C (one middle-temperature cycle: MTC), followed by holding for different periods. Another set of specimens was subjected to cycles between 500 and 740 °C

five times (five LTCs) and then heated to 800 °C, followed by holding for different periods. It should be noted that no AGG occurs at 740 °C. Here, the MTC at 800 °C, but not the HTC at 900 °C, was selected for this examination because the AGG induced by the HTC is too fast to trace the microstructural change. As shown in the optical micrographs in Fig. 4a, b, no AG is seen in the sheet without holding, but abnormally growing grains appear in the samples held at 800 °C. The migration distance of the AGs is plotted in Fig. 4c as a function of annealing time at 800 °C, together with the data of NGG by isothermal annealing at 900 °C[30]. The migration distance of the AGs obtained by isothermal annealing at 800 °C after one MTC is larger by two orders of magnitude than that in the NGG at 900 °C. It is also obvious from Fig. 4c that the migration distance of the AGs in specimens annealed at 800 °C after five LTCs is greater than that in the specimens after one MTC. The GB migration rate after one MTC and five LTCs estimated from the initial slope of each line is $1.6 \times 10^{-5}$ and $8.7 \times 10^{-5}$ m s$^{-1}$, respectively.

The GROD mappings and misorientation angle analyzed by electron backscatter diffraction (EBSD) for one LTC and five LTCs are shown in Fig. 4d, e, respectively. It is found that the orientation spread in grains due to the subgrain microstructure becomes larger with an increasing number of

LTCs. These facts suggest that the sub-boundary energy, depending on the misorientation angle between subgrains, is an important component of the driving pressure in the present AGG. Note that AGG is a phenomenon in which a few large grains rapidly grow in a stagnating fine grained matrix, the GB migration being driven by the capillarity forces due to GB curvature[13]. The grain growth reported in this paper is different from the previously accepted AGG with regard to the driving force. It should also be mentioned that dislocations possibly exist within grains, which can more or less contribute to the driving force of AGG, although it is difficult to quantitatively evaluate it here.

## Discussion

Here, we estimate the growth rate of an AG based on the proposed mechanism and compare it with the experimental results shown in Fig. 4. The growth rate of a grain with radius $R$ is given using GB mobility, $M^{gb}$, and the driving pressure, $\Delta G$, by[15, 31]

$$\frac{dR}{dt} = M^{gb} \cdot \Delta G. \tag{1}$$

The driving pressure for the present AGG is divided into two components: the sub-boundary energy stored in the subgrain structure ($\Delta G_s$) and the driving pressure due to the pre-existing

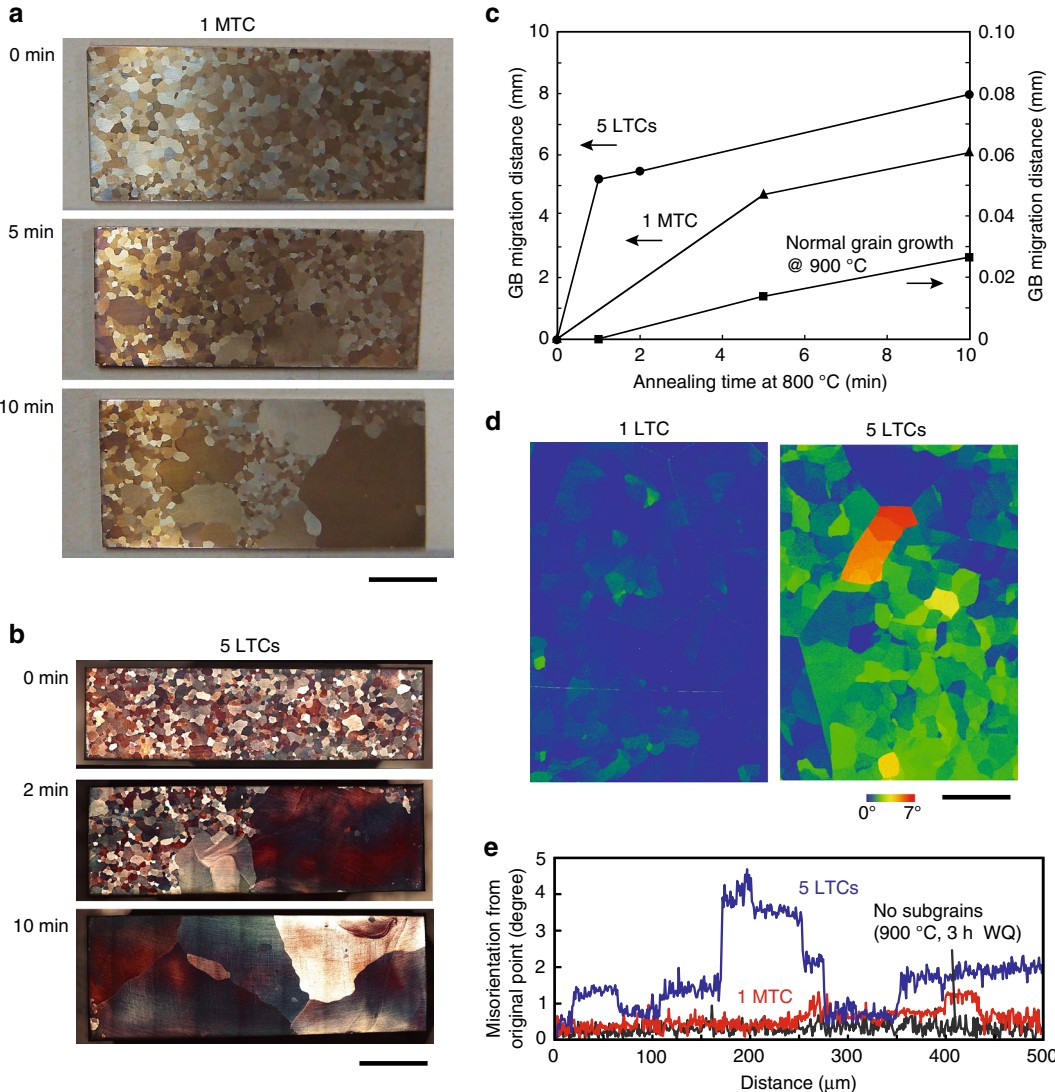

**Fig. 4** Grain boundary (GB) migration distance and subgrain structure. **a** Microstructure of Cu-Al-Mn sheets subjected to one middle-temperature cycle (MTC) (800–500–800 °C) followed by quenching after continuous holding at 800 °C for different periods from 0 to 10 min (Supplementary Fig. 4b). *Scale bar*, 5 mm. **b** Microstructure subjected to five low-temperature cycles (LTCs) (740–500–740 °C) and heated to 800 °C, followed by quenching after holding for different periods from 0 to 10 min (Supplementary Fig. 4c). *Scale bar*, 5 mm. **c** GB migration distance experimentally determined for abnormal grain (AG) shown in Fig. 4a, b obtained after one MTC and five LTCs (Supplementary Fig. 4b, c) as a function of annealing time at 800 °C in Cu-Al-Mn alloy. GB migration distance in the normal grain growth (NGG) mode by isothermal heat treatment at 900 °C is also shown for comparison[31]. **d** Grain reference orientation deviation (GROD) mapping for specimen subjected to one LTC (Supplementary Fig. 4d) and five LTCs (Supplementary Fig. 4e). It is obvious that the orientation mismatch among subgrains after five LTCs is larger than that after one LTC. Scale bar, 200 μm. **e** Misorientation from original point in one LTC (Supplementary Fig. 4d) and five LTC (Supplementary Fig. 4e) specimens of Cu-Al-Mn alloy. Data without subgrains are also shown for reference. The background is about 0.5°. Orientation gaps smaller than 1°, corresponding to a sub-boundary, are detected in one LTC specimen, but gaps up to 2.5° are more clearly observed in five LTC specimen

**Table 1 Constants $C_s$, $C_n$ and $C_a$ in equation (2) for 3D and 2D growth when $R_n \approx R_a$ and $R_n \ll R_a$**

|  | $R_n \approx R_a$ | | | $R_n \ll R_a$ | | |
|---|---|---|---|---|---|---|
|  | $C_s$ | $C_n$ | $C_a$ | $C_s$ | $C_n$ | $C_a$ |
| 3D | 3/2 | 1 | 1 | 3/2 | 3/2 | 1 |
| 2D | 1 | 1/2 | 1/2 | 1 | 1 | 1/2 |

*2D* two-dimensional, *3D* three-dimensional

grains with high-angle boundary ($\Delta G_h$), as given in the following equation:[13]

$$\Delta G_{total} = \Delta G_s + \Delta G_h$$
$$= \frac{C_s \sigma_s V_m}{R_s} + \sigma_h V_m \left( \frac{C_n}{R_n} - \frac{C_a}{R_a} \right), \quad (2)$$

where $\sigma_s$ and $\sigma_h$ are the GB energies of a subgrain and a high-angle pre-existing grain (a NG or an AG) boundaries, $R_s$, $R_n$ and $R_a$ are the mean radii of subgrains and NGs and the radius of an AG, and $V_m$ is the molar volume, respectively. $C_s$, $C_n$ and $C_a$ are constants, depending on the growing dimension, and are listed in Table 1. The $V_m$ of Cu-Al-Mn is $7.6 \times 10^{-6}$ m³ mol⁻¹[32], and $\sigma_h = 0.595$ J m⁻² for Cu-30Zn[33] is used here. The GB energy of a subgrain $\sigma_S$ can be estimated by the following Read–Shockley equation with the misorientation, $\theta$[34]:

$$\sigma_s = \sigma_h \frac{\theta}{\theta_h} \left( 1 - \ln \frac{\theta}{\theta_h} \right), \quad (3)$$

where $\theta_h$ ( $= 15°$) is the critical angle of a low-angle boundary, and at $\theta > \theta_h$ the boundary becomes a high-angle GB with boundary energy, $\sigma_h$ ($=\sigma_s$ at $\theta = 15°$). From the EBSD analysis (Supplementary Fig. 1), the average $\theta$ is 0.46° and 1.12° for one LTC and five LTCs, respectively. Using $\sigma_h = 0.595$ J m⁻² appropriate for Cu-30Zn[33], because of lack of data of Cu-Al-Mn alloy, $\sigma_s$ is calculated to be $8.18 \times 10^{-2}$ J m⁻² after one MTC and $1.60 \times 10^{-1}$ J m⁻² after five LTCs. The $R_s$ can be assumed to be constant ($R_s = 30$ μm for one MTC and $R_S = 24$ μm for five LTCs) because the growth rate is extremely low, as shown in Supplementary Fig 2. While the $R_n$ and $R_a$ are time-dependent, for the initial state ($R_n \cong R_a$), only the first term in Eq. 2 is taken into account and the driving pressure after one MTC and five LTCs are $\Delta G_s = 3.1 \times 10^{-2}$ J mol⁻¹ and $7.6 \times 10^{-2}$ J mol⁻¹, respectively. It is important that this driving pressure due to sub-boundaries hardly decreases by annealing because of the very low growth rate of subgrains (Supplementary Fig. 2). When the AGG progresses, the second term must also be considered, and using $R_n = 400$ μm and $R_a \approx \infty$, the driving pressure after one MTC and that after five LTCs are $\Delta G_{total} = 4.8 \times 10^{-2}$ J mol⁻¹ and $9.3 \times 10^{-2}$ J mol⁻¹, respectively, which are the largest estimations of the driving pressure. The difference in the driving pressure between $\Delta G_s$ and $\Delta G_{total}$ is small (Table 2). This means that the dominant driving pressure for the present AGG phenomenon is the subgrain energy and that the driving pressure is much larger than that of other conventional AGG without subgrains.

The GB mobility is given by the following equation:[31]

$$M^{gb} = \frac{D^{gb}}{\delta RT}, \quad (4)$$

where $D^{gb}$, $\delta$, $R$ and $T$ are the GB diffusion coefficient, the GB thickness, gas constant and temperature, respectively. The $D^{gb}$ is empirically evaluated by the next equation[31] using the melting point, $T_m$:

$$D^{gb} \approx 4 \times 10^{-5} \exp\left[ -\frac{82}{R} \left( \frac{T_m}{T} \right) \right] \quad (5)$$

In the present case, $T_m$ is 948 °C (1221 K), and thus $D^{gb}$ at 800 °C (1073 K) is estimated as being $5.3 \times 10^{-10}$ m² s⁻¹. Using $\delta = 5 \times 10^{-10}$ m[35], $M^{gb}$ is calculated to be $1.2 \times 10^{-4}$ mol m J⁻¹ s⁻¹.

Then the GB migration rate at 800 °C is estimated by Eq. 1 as being $3.7 \times 10^{-6}$–$5.7 \times 10^{-6}$ m s⁻¹ for one MTC and $9.1 \times 10^{-6}$–$1.1 \times 10^{-5}$ m s⁻¹ for five LTCs. In this estimation, the rate of five LTCs is 1.9 – 2.4 times higher than that of one MTC due to higher misorientation of sub-boundaries. A summary of this estimation is listed in Table 2. The growth rate in this estimation is lower than that in the experiments. It is necessary to experimentally obtain detailed data of growth rate, such as by *in situ* observation. Also, accurate parameters for calculation of such factors as the GB diffusion coefficient are necessary. Nevertheless, the growth rate becomes faster in the five cycled specimens due to higher sub-boundary energy shown by calculation and the experimental results agree with this. Thus, it can be concluded that the sub-boundary energy introduced by CHT dominantly contributes to the present AGG and that the LTCs are important for obtaining a ultra-large AG, as shown in Fig. 1b. A similar effect, i.e. increase in driving pressure due to larger misorientation at sub-boundaries, is not obtained by multiple HTCs because AGG starts at temperatures higher than 800 °C in every heating process of HTCs and the sub-boundaries are swept by high-angle GBs. The AGG by LTCs is schematically illustrated in Fig. 3b.

A question may arise as to the subgrains form through the precipitation process. This issue has not been experimentally clarified in this work, but it is believed that the semi-coherency between the matrix and precipitate is related to this phenomenon. It is known that sessile misfit and glissile dislocations exist to accommodate transformation strains of dilatational and shear components, respectively, at the interface between the matrix and precipitate with different structures when they have some specific orientation relationship and when the precipitate grows by the ledge mechanism[36–38]. Such glissile dislocations are probably the source of the sub-boundaries. A large transformation strain is built up when the precipitates grow and the strain should be accommodated by dislocations, resulting in coherency loss. The α precipitate and β matrix have the Kurdjumov–Sachs (K-S), Bain or Pitsch orientation relationship with a semicoherent interface in Cu-Al-Mn[18]. When we compare the GROD mapping of Cu-Al-Mn alloy cooled from 900 to 650 °C and that cooled to 500 °C (Fig. 5), it is seen that the precipitates grow by decreasing temperature and that the orientation deviation becomes remarkable. This fact supports the supposition that the formation of the subgrains is generated through the loss of coherency. Further research is required to reveal the formation process of the subgrains.

Based on the formation mechanism of the AGG, the CHT process shown in Fig. 1a was designed to obtain a large single crystal. The strategy in the heat cycles is as follows: In the initial stage, five HTCs between 900 and 500 °C were performed, where the cooling and heating rate should be enough low (Supplementary Discussion and Supplementary Fig. 3). By this process, a bamboo structure was obtained, but the GBs always remained in the long bars over 300 mm, as shown in Fig. 1d. In the final stage, four LTCs between 740 and 500 °C were applied to obtain a higher driving force, which accelerated the AGG in the bamboo structure. The GB migration rate at 900 °C after five LTCs and the migration distance for 360 min were shown to be $2.2 \times 10^{-5}$–$2.6 \times 10^{-5}$ m s⁻¹ and 466–570 mm, respectively, by calculation. This estimation means that the LTCs can provide a sufficiently high driving force to sweep the remaining GBs

**Table 2 Driving pressures, grain boundary mobility and AGG velocity**

| Heat treatment | $\Delta G_s$ (J mol$^{-1}$) | $\Delta G_h$ (J mol$^{-1}$) | $\Delta G_{total}$ (J mol$^{-1}$) | $M_{gb}$ (mol m J$^{-1}$ s$^{-1}$) | AGG velocity (calc.) (m s$^{-1}$) | AGG velocity (exp.) (m s$^{-1}$) |
|---|---|---|---|---|---|---|
| 1 MTC | $3.1 \times 10^{-2}$ | $1.7 \times 10^{-2}$ | $4.8 \times 10^{-2}$ | $1.2 \times 10^{-4}$ | $3.7 \times 10^{-6}$–$5.7 \times 10^{-6}$ | $1.6 \times 10^{-5}$ |
| 5 LTCs | $7.6 \times 10^{-2}$ | $1.7 \times 10^{-2}$ | $9.3 \times 10^{-2}$ | $1.2 \times 10^{-4}$ | $9.1 \times 10^{-6}$–$1.1 \times 10^{-5}$ | $8.7 \times 10^{-5}$ |

*AG* abnormal growth, *AGG* abnormal grain growth, *LTC* low-temperature cycle, *MTC* middle-temperature cycle
Driving pressures (due to subgrains $\Delta G_s$, pre-existing normal grains with high-angle boundary $\Delta G_h$ and their sum $\Delta G_{total}$), grain boundary mobility of AG $M_{gb}$ and AGG velocities in calculation and experiment for one MTC and five LTC specimens, respectively. $\Delta G_s$ in five LTC specimen is about two times larger than that in one MTC, and its AGG velocity is higher

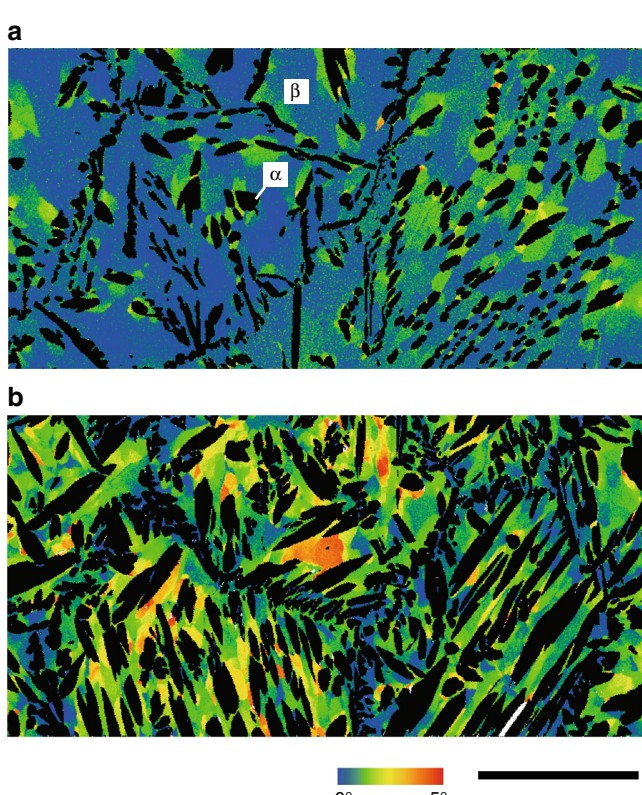

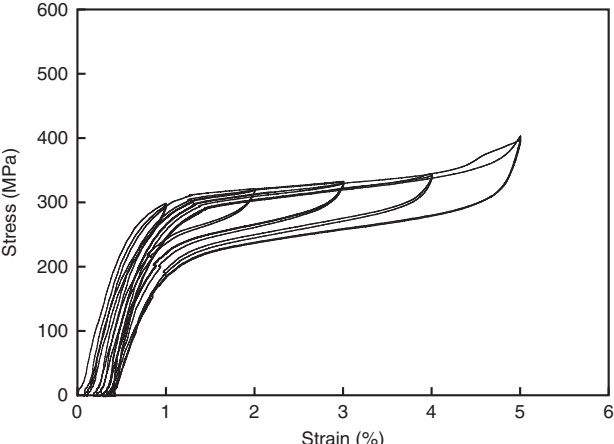

**Fig. 6** Tensile superelastic test in Cu-Al-Mn single crystal bar 15.4 mm in diameter and 682 mm in length. The target strain was incrementally increased up to 5%. Excellent superelasticity was obtained in the long single crystal bar

**Fig. 5** Microstructure of Cu-Al-Mn alloy subjected to slow cooling to the α + β two-phase region. Grain reference orientation deviation (GROD) mapping of Cu-Al-Mn alloy cooled to **a** 650 °C and **b** 500 °C. *Scale bar*, 500 μm. The heat treatment is shown in Supplementary Fig. 4f

out of the long 700 mm bar, leading to a high possibility of creating a single crystal. As a result, single crystal bars 700 mm in length and 15 mm in diameter were obtained, as shown in Fig. 1b. Excellent superelasticity was obtained in the single crystal (Fig. 6 and Supplementary Video 1), while the polycrystalline bar shows large residual strain (Supplementary Video 2).

Since the present technique is advantageous for mass production of single crystals because of the simplicity of the process, this finding opens the way for applications of shape memory single crystals for structural materials, such as for seismic applications in buildings and bridges. Because the AGG related to the subgrain structure has been found in other alloy systems, including in Cu-Zn[39], Fe-Cr-Co-Mo[40–42] and Fe-Mn-Al-Ni[43, 44] alloys, this method of single crystal growth in a solid state can be used with other alloy systems that undergo precipitation with semi-coherency.

## Methods

**Specimen Preparation**. Ingots of $Cu_{71.6}Al_{17}Mn_{11.4}$ were prepared by induction melting in an Ar atmosphere for sheet specimens in an $N_2$ atmosphere for bar specimens. Sheet specimens were obtained by hot-rolling at 800 °C to a thickness

of 2 mm and subsequent cold-rolling to a thickness of 1 mm with intermediate annealing at 600 °C. Bar specimens were obtained by hot forging and cold drawing to a diameter of 15 mm. The β solidus and α solvus temperatures of this alloy were determined to be 948 and 726 °C, respectively, by means of differential scanning calorimetry (DSC). The sheet specimens were solution-treated at 740–900 °C for 5–15 min in the β single-phase region. They were cooled to 500 °C in the α + β two-phase region at a cooling rate of 3.3 °C min$^{-1}$ and held for 10 min at 500 °C, and then heated to 760–900 °C at a heating rate of 10 °C min$^{-1}$ and held for various periods, followed by quenching in water. Some specimens were subjected to CHT between 740 and 500 °C five times, instead of the single CHT, before heating to the β single-phase region. The procedure of heat treatment of the bars and sheets is illustrated in Fig. 1 and Supplementary Fig. 4.

**Microstructural observation**. The microstructure and the crystallographic features were investigated by optical microscopy and EBSD using a field emission scanning electron microscope. Method of determination of misorientation angle between subgrains is shown in Supplementary Fig. 1, and the accuracy of misorientation angle[45, 46] is discussed in Supplementary Discussion.

**Evaluation of grain size**. The grain size of NGs and subgrains was evaluated by the linear intercept method using the optical micrographs. The length of a line segment drawn on the optical microscopy images, $L$, is described as follows:

$$L = N\bar{l}, \qquad (6)$$

where $N$ is the number of grains on the line segment and $\bar{l}$ is the average length of one grain. In three-dimensional (3D) grain growth[13], a grain is taken as a sphere with a mean radius of $\bar{R}$ and the $\bar{l}$ is given by

$$\bar{l} = \frac{4}{3}\bar{R}, \qquad (7)$$

Thus, the average grain radius $\bar{R}$ is described using the Eqs. (6) and (7) as

$$\bar{R}_{3D} = \frac{3L}{4N} \qquad (8)$$

**Measurement of migration distance of grain boundaries**. The sheet specimens for determination of the GB migration distance for AGs were subjected to CHT using quartz capsules backfilled with Ar and quenched in water after holding for various periods at 800 °C. For the measurement of migration distance, the specimens etched after electropolishing were used and the GBs and sub-boundaries were observed using optical microscopy. Several AGs were observed in one specimen, but some grains probably started to grow after incubation. Therefore, the maximum width of the subgrain-free zone, i.e., the maximum distance between the subgrain region and the high-angle boundary, was defined as the migration distance. Once a growing AG faces another AG, the growth mode becomes NGG and the rate of GB migration becomes slow because the energy of subgrain boundaries has already been consumed. Therefore, the maximum migration distances of GBs of AGs that face subgrains were measured.

**Superelastic tests**. The superelasticity of Cu-Al-Mn single crystal bar 15.4 mm in diameter and 682 mm in length was evaluated by a tensile test at room temperature. The gauge length was 400 mm and the strain rate was $1.7 \times 10^{-4}\,\mathrm{s}^{-1}$. The bar was first loaded to 1% strain and unloaded, and then the target strain was increased in 1% increments, each cycle being repeated twice.

**Data availability**. All relevant data are available from the corresponding author upon request.

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

## Acknowledgements

This work was supported by the A-STEP program from the Japan Science and Technology Agency (JST) and by a Grant-in-Aid for Scientific Research from the Japan Society for the Promotion of Science (JSPS). We thank H. Kokawa, T. Furuhara, G. Miyamoto, K. Ishikawa and M. Fujii for helpful discussions. We also thank N. Yoshida for his invaluable assistance with the mechanical test.

## Author contributions

T.K. determined the growth rate of abnormal grains and performed the EBSD analysis. T.S. determined the growth rate of subgrains. S.K. and T.T. carried out the heat treatment of the bars and the superelastic bending test. Y.A. performed the superelastic tensile test.

T.K., T.O. and R.K. interpreted the data and wrote the manuscript. All the authors discussed the results and approved the manuscript.

## Additional information

**Competing interests:** The authors declare no competing financial interests.

