## [Peer Review File · Nature Communications]

This manuscript has been previously reviewed at another journal that is not operating a transparent peer review scheme. This document only contains reviewer comments and rebuttal letters for versions considered at Nature Communications.

Reviewers' Comments:

Reviewer #1:

Remarks to the Author:

The authors demonstrate a new type of abnormal grain growth (AGG) in metallic alloys, building on their earlier discovery, based on the subgrain boundary formation in high volume fractions. They use this mechanism as an alternative to known single crystal growth techniques for metals and fabricate single crystal samples as large as 70 cm in length. They demonstrate this AGG mechanism in a CuMnAl shape memory alloy taking advantage of subgrain formation due to the diffusional transformation between two phases at high temperatures. As compared to their previous work published in Science, they show they can grow the grains into much larger diameters leading to these extremely large single crystals by increasing the stored energy at subgrain boundaries by thermal cycling between two phase and single phase regions without heating the samples to very high temperatures not to allow AGG initially, then heating to sample to high temperatures to allow AGG with high migration speeds. In addition, they develop simple analysis approach to predict the grain growth rates and demonstrate that their calculations more or less follow the trends observed experimentally.

This is an exciting method that will open a new research area to grow metallic single crystals relatively easier than known melt base single crystal growth methods, and with much higher sample yields. In addition, the physical mechanisms behind this AGG will need to be further investigated and feasibility for other metals with two phase regions will need to be revealed. Therefore, the present manuscript could be published in nature communications as is.

Response to Referee

We sincerely thank the reviewer once again for his/her effort to review our manuscript and for positive and insightful comments. We feel that the corrections have much improved our manuscript.

REVIEWER'S COMMENTS:

The authors demonstrate a new type of abnormal grain growth (AGG) in metallic alloys, building on their earlier discovery, based on the subgrain boundary formation in high volume fractions. They use this mechanism as an alternative to known single crystal growth techniques for metals and fabricate single crystal samples as large as 70 cm in length. They demonstrate this AGG mechanism in a CuMnAl shape memory alloy taking advantage of subgrain formation due to the diffusional transformation between two phases at high temperatures. As compared to their previous work published in Science, they show they can grow the grains into much larger diameters leading to these extremely large single crystals by increasing the stored energy at subgrain boundaries by thermal cycling between two phase and single phase regions without heating the samples to very high temperatures not to allow AGG initially, then heating to sample to high temperatures to allow AGG with high migration speeds. In addition, they develop simple analysis approach to predict the grain growth rates and demonstrate that their calculations more or less follow the trends observed experimentally.

This is an exciting method that will open a new research area to grow metallic single crystals relatively easier than known melt base single crystal growth methods, and with much higher sample yields. In addition, the physical mechanisms behind this AGG will need to be further investigated and feasibility for other metals with two phase regions will need to be revealed. Therefore, the present manuscript could be published in nature communications as is.